# Periodontal Therapy in Bariatric Surgery Patients with Periodontitis: Randomized Control Clinical Trial

**DOI:** 10.3390/jcm11226837

**Published:** 2022-11-19

**Authors:** Dejana Čolak, Alja Cmok Kučič, Tadeja Pintar, Rok Gašperšič

**Affiliations:** 1Department of Oral Diseases and Periodontology, Dental Clinic, University Medical Centre Ljubljana, 1000 Ljubljana, Slovenia; 2Faculty of Medicine, University of Ljubljana, 1000 Ljubljana, Slovenia; 3Department of Abdominal Surgery, University Medical Centre Ljubljana, 1000 Ljubljana, Slovenia

**Keywords:** bariatric surgery, periodontitis, non-surgical periodontal therapy, morbidly obese, weight loss surgery, bariatric surgery complications, periodontal intervention, oral health

## Abstract

Background: Bariatric surgery (BS) patients may experience the progression of periodontitis during recovery. We aimed to determine whether non-surgical periodontal therapy before BS improves the periodontal and systemic health parameters after the surgery. Methods: BS candidates with periodontitis were randomized into the test (TG) and control group (CG). One month before BS (pre-BS), patients in the TG (*n* = 15) received non-surgical periodontal therapy, while patients in the CG (*n* = 15) received only mechanical plaque removal. Patients were re-examined 3 and 6 months after BS. Differences between the TG and CG in clinical periodontal parameters, systemic health-related serum biomarkers, parameters of obesity, and prevalence of obesity-related diseases were evaluated. Results: From the 30 included patients, 26 were re-examined at 3 months and 20 patients at 6 months. Periodontal parameters bleeding on probing (*p* = 0.015), periodontal pocket dept (PPD, *p* = 0.0015), % PPD > 4 mm (*p* < 0.001), and full-mouth plaque levels (*p* = 0.002) were lower in the TG than in the CG at 6 months after BS. There is a general improvement in systemic health after BS without significant differences (*p* > 0.05) between the TG and CG at the 6-month follow-up. The TG shows a tendency for improvement in metabolic syndrome components at the 6-month follow-up compared to pre-BS (*p* < 0.05). Conclusions: Non-surgical periodontal therapy in periodontitis patients before the BS may improve periodontal health 3 and 6 months after the surgery. The possible benefits of periodontal therapy on the overall health of BS patients should be further explored.

## 1. Introduction

The obesity epidemic started more than 40 years ago and is still ongoing [1,2]. The list of diseases and conditions connected to obesity is long and includes systemic inflammation [3,4], diabetes, hypertension, metabolic syndrome (MS), metabolic-associated fatty liver disease, as well as oral health problems [5,6,7]. According to estimates, patients with obesity have worse oral health than the overall population [8].

Periodontitis is a chronic inflammatory progressive disease of the tooth-supporting tissues [9], more common in patients with obesity than in normal-weight individuals [7,10,11], and being recognized as a manifestation of obesity at the 2017 World Workshop [12]. Periodontitis originates from the interaction between bacterial dysbiosis and a modulated host immune response [13,14,15]. Oral and periodontal health in patients with obesity is affected by obesity-related diseases such as diabetes [16], MS [17], anemia [18,19], depression [20], eating disorders [21,22], gastroesophageal reflux disease (GERD), as well as high calorie-low fiber food diet [23,24], and avoidance of medical interventions due to weight-stigma in medical health professionals [25,26]. Diabetes is recognized as a risk factor influencing the progression rate of periodontitis, modifying the periodontitis grade to a worse score, according to the AAP/EPF classification criteria [27,28,29]. In patients with obesity, the obesity-induced proinflammatory state is marked even in the gingiva of periodontally-healthy patients [30,31,32]. This predisposes patients to more significant periodontitis involvement and loss of periodontal tissues [33]. At the same time, adolescents and adults with elevated BMI have more frequently poor oral hygiene, further promoting periodontitis [32]. Periodontitis has already been independently linked to many systemic diseases [34], including MS [35], cardiovascular disease [36,37], diabetes [38], premature births [39], neurological diseases [40], and metabolic-associated liver disease [41,42,43]. Current findings indicate a contributory effect of periodontitis in many obesity-driven diseases [44]. Bacteria by-products and inflammatory mediators caused by periodontitis can seep into circulation and affect distant locations [45,46]. In prospective studies, the presence of periodontitis has been connected to the future onset of metabolic syndrome [47] and cardiac events [48]. To further support these claims, successful periodontal therapy has been connected to lower markers of systemic inflammation [49,50,51] and improvement in systemic health parameters in patients with obesity-related diseases such as cardiovascular diseases [50,52], diabetes [53], and dyslipidemia [54]. Non-surgical periodontal therapy in patients with MS has been shown to improve blood pressure, glucose levels, and serum markers of systemic inflammation [49]. Periodontal treatment in patients with obesity can improve the levels of glucagon-like peptide 1 [55] and other parameters of diabetes [56]. However, researchers speculate that the presence of obesity might reduce the success of periodontal therapy [57,58]. On the other hand, the addition of conventional weight loss to periodontal therapy in patients with obesity might restore the beneficial effect of periodontal treatment [59].

Bariatric surgery (BS) is a broad term for describing surgical alterations of patients’ digestive systems to achieve weight loss and resolve of obesity-related comorbidities [60,61]. The main principle of BS lies in the reduction of stomach size and alteration of the food path through the small intestine. This leads to restricted food intake and a decrease in the absorption of nutrients. The surgery is indicated for patients with a body mass index (BMI) > 40 kg/m^2^, or less if the obesity-related disease persists [60,62]. Lifestyle changes are mandatory for achieving the best long-term results [60]. BS patients may expect to lose around 30% of weight after a year [63] and resolution of obesity-related comorbidities between 50 and 90% [63,64]. However, oral health deteriorates after BS [65,66,67]. The epidemiological studies showed periodontal health deterioration after BS [68]. The cohort and cross-sectional studies on BS patients showed that patients experienced worse clinical periodontal parameters 6 to 12 months after BS than before surgery [67,69,70] and a shift in oral microbiome towards increased numbers of periodontal pathogens after the BS [70,71]. Furthermore, radiographically diagnosed alveolar bone loss and sparser trabecular pattern of mandibula were found in patients 6 months after BS [72]. This is unanticipated as BS neutralizes some of the risk factors for poor oral health due to weight loss, lowering of systemic inflammation, and resolution of diabetes [59]. Meanwhile, oral complications after BS may occur due to systemic complications (gastric reflux [66], vomiting [65], anemia, change in the gut microbiome [73], and malnutrition [74,75]) and a change in diet after BS [65,76,77]. Weight regaining after BS and unresolved eating disorders can also negatively affect oral health [21,22]. During recovery from BS, a soft-food diet and frequent meals are likely to encourage dental plaque formation [60,78,79]. Some patients struggle to follow post-operative dietary advice. They may also adopt some unhealthy habits such as restrained eating, nonvitamin use, binge eating, and consuming high-calorie soft/meltable foods [80,81,82]. Bone metabolism can be affected in BS patients due to malabsorption of calcium and vitamins C, K, B, and D [75]. Malnutrition is linked to osteoporosis and bone loss in periodontitis-affected teeth [72,75]. The systemic BS complications might be mitigated with appropriate post-operative recommendations and regular check-ups [60].

In adults, periodontitis is still a key factor for tooth loss, which reduces masticatory function [23,24,83,84]. Even though the overall quality of life after BS improves, studies in BS patients reveal that it has a detrimental effect on oral health-related quality of life (OHRQoL) [85,86,87]. On the other hand, no protocol for BS implements dental preparations of patients with obesity [88], even though these patients suffer from poor oral health [8,60,89]. A study showed that oral-hygiene instructions and diet recommendations before BS (pre-BS) may have a beneficial effect on periodontal tissues [90].

The combined adverse effect of obesity and BS on patients’ periodontal health are evident [33,68]. So far, no randomized control trial (RCT) with periodontal intervention before BS has been conducted to improve oral health and assess its effects on general health and well-being.

The study aims to determine whether non-surgical periodontal therapy in conjunction with BS improves periodontal and systemic health during the recovery from the surgery.

## 2. Materials and Methods

### 2.1. Study Design

The study was randomized, controlled, parallel-group, and double-blinded (patients and the examiner). We tested the effects of non-surgical periodontal therapy one month before BS in candidates with periodontitis on their periodontal and systemic health parameters 3 and 6 months after BS. For this purpose, previously recruited patients with periodontitis were randomly allocated into test (TG) and control groups (CG). Patients received periodontal therapy according to allocation one month before BS, i.e., patients in the TG received non-surgical periodontal therapy and those in the CG received minimal periodontal treatment, similar to placebo treatment described by Montero et al. [49]. A detailed explanation of the periodontal therapies performed in the TG and CG is presented in the section “Periodontal therapy in test and control groups”. In addition, all patients in the study received oral hygiene education and motivation at baseline (pre-BS) and follow-ups (3 and 6 months). No additional periodontal therapies were performed at the follow-ups.

The CONSORT statement was implemented while reporting the design, analysis, and interpretations of the study results. The study protocol followed the Declaration of Helsinki and was approved by the Republic of Slovenia’s National Medical Ethics Committee (0120-312202010). All patients received verbal and written explanations and gave their written consent to participate in the study. The study protocol was registered at clinical trials as NCT04653714 [91].

### 2.2. Patients’ Inclusion Criteria for the Studies

Patients with severe obesity, ≥18 years old, and with the indication for BS treated at the Department of Abdominal Surgery, University Medical Centre (UMC), Ljubljana, Slovenia, were referred to be included in the studies. The indication for BS was made by the experienced bariatric surgeon (TP) following the published guidelines [60]. BS was indicated if a patent had a BMI > 40 kg/m^2^ or BMI > 35 kg/m^2^ and obesity-related comorbidity [60]. At UMC Ljubljana, Slovenia, the following BS surgeries that were performed were laparoscopic: one-anastomosis gastric bypass (OAGB), Roux-en-Y Gastric Bypass (RYGB), and sleeve gastrectomy. The bypass surgeries, i.e., OAGB and RYGB, were made with a standard limb of 150 cm [92,93].

BS applicants underwent a detailed dental and periodontal examination at the Department of Oral Medicine and Periodontology, UMC, Ljubljana, Slovenia. BS candidates were included in the RCT if they were diagnosed with periodontitis (AAP/EPF classification) [27]. To be categorized as periodontitis, patients should exhibit detectable ≥ 1 mm interdental CAL or buccal/oral CAL of 3 mm with PPD of >3 mm on two or more non-adjacent teeth [27,28]. Based on the clinical, radiological evaluation, and systemic health status, periodontitis patients were further categorized by the staging (I-IV) and grading system (A, B, and C) [27,28].

Patients were excluded if they refused to participate, if they had fewer than 16 teeth, pregnant or lactating, patients with severe psychiatric disorders, severe illnesses not related to obesity, a medical history of malignant disease less than 5 years ago, and if they had contraindications for BS (e.g., severe CVD, medical and other general contraindications for surgery regarding the method of treatment and postoperative monitoring, previous multiple abdominal surgeries).

### 2.3. Sample Size Calculation

For the study, the sample size was calculated based on the data from previous studies on patients with obesity [94] with PS software [95]. The sample size was calculated based on detecting a mean difference in PPD between TG and CG of 0.6 mm with an estimated standard deviation (SD) of 0.6 mm. A minimum of 14 subjects per group is required to detect statistically significant differences (α = 0.05) with an 80% power. In the end, 15 subjects were included in each group (TG *n* = 15; CG *n* = 15). Initially, more significant number of patients were planned to be included in the study (20 subjects in each group) to account for the dropouts. However, due to COVID-19 epidemic restrictions, they were not able to be recruited.

### 2.4. Patient Recruitment

The randomized clinical trial was conducted from September 2020 until July 2022 following the flowchart in Figure 1. Due to obstacles presented by the COVID-19 epidemic, the study was completed later than initially expected. A total of 81 patients were referred to the Dental Clinic for periodontal examination upon setting the indication for BS at the Department for Abdominal Surgery. Upon periodontal examination, 3 patients were excluded from participating in the RCT study as they were periodontally healthy, 2 patients were edentulous, 8 patients had less than 16 teeth present, 2 had severe diseases not related to obesity, 2 refused to participate, 5 underwent BS before the study began, 9 abended pursue of BS, while 20 patients were diagnosed with gingivitis and included in another study under the same protocol (NCT04653714) [91]. Of the remaining individuals, 30 were diagnosed with periodontitis (AAP/EPF classification) [27] and were assigned to the TG or CG according to randomization. The patients’ randomization was carried out using simple randomizations by creating random allocation sequence numbers using a computer software algorithm (DČ). This resulted in fifteen patients in each group, i.e., TG = 15 and CG = 15 patients. The patients were recalled for periodontal therapy allocated from randomization sequence around 4 weeks prior to BS. From those, all 30 patients underwent BS and periodontal therapy.

### 2.5. Dental and Periodontal Clinical Examinations

BS candidates were referred by the BS surgeon (TP) to the Dental Clinic, Department of Oral Diseases and Periodontology, for dental and periodontal examination. Patients had detailed dental and periodontal examinations before surgery and at follow-ups (3- and 6-month follow-ups). The same clinical protocol described in a previously published article was followed [89]. In brief: with a mm periodontal probe (PQW Williams, Hu-Friedy, Chicago, Illinois, USA), a blinded and calibrated examiner (RG) recorded the following periodontal parameters on each existing tooth, excluding third molars, at 6 sites: full-mouth plaque index (FMPI, % site) [96], probing pocket depth (PPD, mm), gingival recession (REC, mm). From PPD and REC, clinical attachment loss (CAL, mm) was computed after the fact. Radiography analyses were done by the OPT. The examiners’ (RG) intraclass correlation coefficients for PPD and CAL above were 0.9 and kappa values for Pl, GBI, and BOP were above 0.95 [97]. The AAP/EFP classification was used to determine the correct periodontal diagnosis, and stage and grade of disease [27,98].

Patients provided the following responses to an interview question about their habits: smoking (no/yes), alcohol consumption (more/less than 12 alcohol units a month), regular weekly exercise (yes/no), daily oral hygiene (using of a toothbrush, toothpaste, and interdental hygiene tools), and regular twice a year dental check-ups (yes/no) [89]. The OHIP-14 self-filed questionnaire was used to evaluate the change in OHRQoL in BS patients following the intervention [99]. The OHIP-14 questionnaire was translated into the Slovenian language and verified [100]. The questionnaire consisted of 14 questions, divided into 7 domains of impact (2 questions for each domain): functional limitation, physical pain, psychological discomfort, physical disability, psychological disability, social disability, and handicap [99]. The higher the total OHIP-14 score (0–49), the more decisive on the OHRQoL was noticed [99].

### 2.6. Systemic Parameters

Bariatric surgeon (TP) before the surgery and at controls referred patients to a medical specialist for evaluation to assess the presence, extent, and treatment of obesity-related comorbidities (diabetes, hypertension, dyslipidemia, obstructive sleep apnea, polycystic ovary syndrome, spine and joint problems due to excess weight) as described in detail in our previous study on BS candidates [89]. Diagnoses of GERD and MS were assessed and treated by a gastroenterologist (TP) [101]. Data on the patient’s demographic (sex, age, level of education), anthropometric data (weight, height, waist size), and medical history were obtained from the medical records. The body mass index (BMI; kg/m^2^) was calculated using anthropometric data on weight and height, whereas the Edmonton Obesity Staging System (EOSS) was based on available medical history [102]. Before the BS and at the 6-month follow-up, serum levels of C-reactive protein (CRP), aspartate transaminase (AST), alanine transaminase (ALT), GGT, albumin, triglyceride, high-density lipoprotein (HDL), low-density lipoprotein (LDL) cholesterol, glucose and glycated hemoglobin (HBA1c) were measured [103,104].

### 2.7. Periodontal Therapy in Test and Control Groups

Around 4 weeks prior to BS, patients were recalled for baseline examination by blinded experienced (RG) and periodontal therapy according to the randomized allocation done by an experienced therapist (ACK). All patients, regardless of allocation, received motivation and instruction in proper oral hygiene, which was reinforced at each visit. Subjects in TG received non-surgical periodontal therapy, i.e., the removal of supra- and subgingival deposits using piezoelectric ultrasonic instruments (PiezoLED ultrasonic scaler with Piezo Scaler tip 203 (KaVo dental, Biberach, Germany)), followed by scaling and root planning of sites with PD ≥ 5 mm under local anesthesia (Ultracain©, Hoechst, France) using Gracey curettes (Hu-Friedy, USA) and polishing with mechanical brush and professional toothpaste (Proxyt RDA 7 Ivoclar Vivadent). CG subjects received low-intensive supragingival plaque removal with a mechanical brush and professional toothpaste (Proxyt RDA 7 Ivoclar Vivadent). At 3 and 6 months after BS, patients we recalled for periodontal evaluation at the Dental Clinic and for evaluating systemic health at the Department of Abdominal Surgery. At the end of the study period, all patients in CG received non-surgical periodontal therapy, while patients in TG had repeated periodontal therapy or were referred for periodontal surgical therapy as necessary.

### 2.8. Statistical Analysis

Descriptive statistics was used to present the data on periodontal health, systemic health, demographic data, and behavioral patterns. If quantitative data indicated a proclivity for normal distribution, the mean (standard deviation) was used; otherwise, data were displayed as the median (25–75 quartal). The study’s primary outcome was the difference in periodontal clinical parameters between groups after BS. Secondary outcomes were differences in OHRQoL and obesity-related systemic parameters (BMI, obesity-related comorbidities, and serum biomarkers). With Welch’s one-way ANOVA test, we tested the difference in clinical periodontal parameters, OHIP-14, and BMI between the periodontitis TG and CG for measurements at pre-BS and control (3, 6 months). The Mann–Whitney U-test (two-tailed) was applied to test differences within and between groups for numerical variables. Fisher’s exact test was used to determine the difference in categorical variables within and between-groups at each follow-up. The significance threshold was set at alpha 0.05. Data analysis was conducted in R (R Core Team 2020) [105] and Microsoft Excel [106].

## 3. Results

### 3.1. Patients’ Baseline Characteristics

From thirty (*n* = 30) treated patients, twenty-six (*n* = 26) were re-examined after 3 months, twenty (*n* = 20) after 6 months from BS, and others were lost during follow-up. Only data from the re-examined patients were presented in the results. Patients in the study did not experience serious complications from BS or periodontal interventions. The majority of the pre-BS characteristics of patients in the TG and CG before BS are shown in Table 1. There were no significant differences between the TG (*n* = 14) and CG (*n* = 12, *p* > 0.05) pre-BS. Most of the patients underwent OAGB (61.5%, 16/26), while others underwent sleeve gastrectomy (23%, 6/26) and RYGB (15%, 4/26). All procedures were done laparoscopically. Patients were 50.8 (SD = 9) years old on average. Around 70% of the patients in both groups were women. Their BMI was 48 (6.3) kg/m^2^, EOSS stage 3 was the most prevalent, obesity-related comorbidities prevalent between 14–86%. Half of the patients reported the reason for BS was to lose weight, while the other half reported that the reason was to improve health. Around half (TG: 50%, CG: 42%) of participants had higher education, 36% in the TG and 25% in the CG reported they were smokers, 7% in the TG and 8% in the CG consumed more than 12 units of alcohol a month. More than half of the patients (TG: 64%, CG: 58%) reported they visited the dentist twice a year, while less than half (TG: 43%, CG: 33%) reported they properly maintained oral hygiene (Table 1).

### 3.2. Changes in Test and Control Group after Bariatric Surgery

After BS, all patients in the study experienced significant weight loss without differences between the TG and CG (Table 2).

#### 3.2.1. Change in Periodontal Parameters after Bariatric Surgery

At the 3-month follow-up, TGs showed significant improvement in PPD and % PPD > 4 mm compared to the CG (*p* < 0.05). At the 6-month follow-up, the TG showed a significant difference in BOP, PPD, % PPD > 4 mm compared to the CG (*p* < 0.05). Compared to measurements pre-BS, only the TG showed improvement in PPD > 4 mm (%) and plaque (FMPI) levels at 3- and 6-month follow-ups. In the TG, CAL and REC stayed similar to the beginning of the study, while the CG showed a tendency for worsening in CAL at 6 months (*p* > 0.5). Generally, OHIP-14 scores were low both pre- and post-BS (Table 3). After BS, there was no significant difference between the TG and CG in OHIP-14 score. However, the TG showed a slight improvement compared to the CG (OHIP-14: 6-month follow-up TG: 4 (2–9), CG: 15 (4–17), *p* > 0.05; Table 3). Both groups’ oral hygiene habits improved after BS (*p* = 0.09, Table 3).

#### 3.2.2. Change in Obesity-Related Comorbidities after Bariatric Surgery

There was a resolution of obesity-related comorbidities 3 and 6 months after BS, in both the TG and CG. In a short period of half a year after BS, the most significant improvement can be seen in sleep apnea (pre-BS TG: 79%, CG: 58%; 6-month: TG: 9%, CG: 11%, *p* < 0.05, Table 4), while depression seems to improve the least (pre-BS TG: 14%, CG: 17%; 6-month: TG: 18%, CG: 22%, Table 4).

There is an overall significant change after BS in BMI, MS, diabetes, high cholesterol, and sleep apnea (*p* < 0.05; Table 4). There was no significant difference in the prevalence of obesity-related comorbidities between the TG and CG 6 months after BS (*p* > 0.05). Six months after BS in the TG, when compared to pre-BS, there was a significant drop in the prevalence of MS, diabetes, high cholesterol, hypertension, and joint and muscle problems (*p* < 0.05, Table 4). In the CG, there was only a significant drop in the prevalence of diabetes and sleep apnea 6 months after BS (*p* < 0.05, Table 4) compared to pre-BS.

The prevalence of reported smokers and alcohol consumption decreased after BS, while there was a reported increase in regular physical activity (*p* > 0.05, Table 4).

#### 3.2.3. Systemic Biochemical Serum Parameters before and after Bariatric Surgery

Biochemical analysis of serum 6 months after BS showed improvement in most obesity-related parameters (CRP, GGT, LDL, triglyceride, glucose, HBA1c, Table 5). However, a more significant improvement was expected in HDL. At 6 months after BS, there was no significant difference between the TG and CG in biochemical serum analysis (Table 5). Both the TG and CG showed significant lowering of CRP levels (Pre-BS TG: 12 (8–12.5), CG: 16 (11–21); 6-month follow-up TG: 3 (2.3–6.3), CG: 4.5 (2.8–9.3) *p* < 0.05). The TG at 6 months after the surgery showed a significant drop in albumin levels (*p* < 0.05), and a tendency for lower glucose levels (*p* = 0.09, Table 5) compared to pre-BS.

#### 3.2.4. Dietary Regime after Bariatric Surgery and Related Complications

The dietary regime after bariatric surgery and related complications did not differ between the TG and CG. Almost all patients reported taking supplements recommended by surgent, i.e., Bariemon multivitamins and minerals (Enemon, Slovenia), D-vitamins, and calcium citrate after BS (Table 6). Patients reported that they had five to six smaller meals a day. After BS, patients were on average 4.4 (SD = 1) weeks on a liquid diet, followed by 3 (SD = 1.8) weeks on a soft-food diet, and most of the patients were on a solid diet at 3-month control. Vomiting after BS was reported by 36% of patients, which lasted on average 3.5 (1.2) weeks after BS (Table 6).

## 4. Discussion

The study showed that the non-surgical periodontal therapy in periodontitis patients one month before BS can improve patients’ periodontal health after the surgery. The pre-operative periodontal therapy and proper oral hygiene education and motivation might neutralize periodontal deterioration after BS [68]. The effects of periodontitis therapy on systemic health after BS remain questionable. However, patients in the non-surgical periodontal therapy group tended to have higher resolution of MS components than patients who only received low-intensive supragingival plaque removal. Nevertheless, further research is needed to test the previously mentioned observation.

Our study is the first RCT to explore the effects of nonsurgical therapy in BS patients. The study combined periodontal healing with weight loss due to BS in patients with class III obesity [59]. Even though obesity is known to lower the effects of periodontal intervention [57], the beneficial effects of non-surgical periodontal therapy on the periodontal health of patients in TG were quite clear. In a similar study, supragingival scaling and diet advice to BS candidates with periodontitis improved BOP, gingival and plaque indices but not PPD and CAL 6 months after BS [90]. Compared to the previously mentioned study, patients in the TG also received removal of subgingival deposits, which might be the reason for the more significant improvement in their periodontal health. In the RCT by Porcelli et al. [107], the effect of intense oral health promotion on the periodontal health of BS patients was tested. The results showed improvement in dental plaque levels and community periodontal index score 6 months after BS compared to pre-BS. This study’s results point to the importance of proper education and motivation of BS candidates, which was also confirmed in our study.

The slight improvement in OHRQoL in the TG after BS suggests the favorable effect of periodontal therapy on patient’s well-being [108]. A more significant improvement in OHRQoL after BS was expected in the TG because of periodontal therapy [109]. However, this might not have occurred due to other oral health-related problems associated with obesity, such as carious lesions, enamel erosions, tooth loss, and inadequate rehabilitation [110,111,112]. These issues were not addressed by periodontal therapy. This points to a multidisciplinary approach that is needed in managing the oral health of bariatric candidates. In the CG, periodontal parameters after BS stayed similar to pre-BS, without expected worsening [68]. This might be due to low-intensity periodontal therapy in CG, which was conducted to conceal the randomization allocation from the patients. The absence of serious periodontal worsening among study patients can be attributed to education and motivation on proper oral hygiene [90]. Another critical element might be the applied efforts to reduce the incidence of BS complications that may impact oral health [60]. All the patients in our study were recommended to take daily supplements of minerals and vitamins for BS patients as suggested by the guidelines [60]. To prevent issues with the stomach post-operatively and consequently acid reflux and GERD, patients were prescribed proton pump inhibitors which neutralize stomach acid for up to 3 months, or longer if needed [60]. A study that followed BS patients without dental intervention revealed the absence of periodontal worsening after BS [113]. One explanation for this might be the patients’ higher socioeconomic position and educational levels [114], which meant patients were more likely to be already educated about oral hygiene prior to BS [113]. We believe education and management of BS patients are needed to lessen the occurrence of systemic complications and their effect on oral health.

The effects of BS on weight loss, resolution of obesity-related comorbidities, and change in serum biochemical parameters in both groups were as expected for a period of 3 to 6 months after BS [115,116,117,118]. Both groups have almost complete remission of diabetes and remaining sequels of obstructive sleep apnea. The lowered CRP levels showed a decrease in systemic inflammation in both groups after BS. In addition, there is a tendency for improvement in parameters of liver health in both groups after BS (levels of GGT, LDL, triglyceride) and parameters for diabetes (glucose and HBA1c levels), while there was a significant decrease in albumin levels in the TG compared to pre-BS.

Six months after BS, there was no observable difference in the prevalence of obesity-associated comorbidities between the TG and CG. Yet, there was a trend for reduction in MS in the TG 6 months after BS compared to pre-BS, but not in the CG. The prevalence of individuals with elevated cholesterol and hypertension in the TG decreased more than in the CG, compared to pre-BS. However, the prevalence of MS and its components in both groups are still consistent with previous research studies on disease resolution after BS [64,119]. Over the next two years, more remarkable improvement in MS and other obesity-related comorbidities is anticipated [64,117]. Nevertheless, given that periodontitis is associated with obesity-related comorbidities such as MS and diabetes, research into how periodontal therapy affects these conditions is essential [35,38,120]. Other interventional studies have shown that periodontitis therapy can have beneficial effects on systemic health parameters in average weight or patients with obesity, such as lowering blood pressure [50,121], improving diastolic cardiac function [122], endothelial dysfunction [52,123], inflammatory markers (CRP, TNF α, IL1-b) [49,50,55], lipid profile [54], diabetes parameter [49] and improves glycemic control [53]. Due to the limitations of our modest-sized short prospective study, we cannot confirm a link between periodontal therapy and the resolution of MS and its components. The suggested discrepancies between the TG and CG in the prevalence of systemic diseases could be explained by other variables unrelated to periodontal therapy. After BS, the TG reported a higher prevalence of physical activity and lower serum albumin levels, which may indicate a more significant caloric deficit in the TG. [124]. At 6 months post-BS, the patients in the TG experienced fewer joint and muscle problems compared to pre-BS. Those mentioned above could have encouraged physical activity and, therefore, faster resolution of MS components. Future research on BS patients and the effects of periodontal intervention should investigate these findings in more detail. The more significant impact of obesity may overshadow the negative effects of periodontitis on the systemic health of patients with obesity. However, some research shows that the periodontitis–systemic health association exists even in the presence of obesity [125,126].

The main limitation of our study is a modest sample size at the follow-up. Increasing the sample size in future studies would give us a better overall influence of non-surgical periodontal therapy on the periodontal and systemic health of BS patients. The short follow-up period of our study also lingers the evidence of potential beneficial effects of periodontal therapy on systemic health in BS patients. BS patients will experience improvement in systemic health in a year to two years after BS [60]. In the trial, we anticipated a more considerable improvement in some metrics, such as the lipogram, but this could occur in the following months after the study period. Patients in the CG received a low-intensity periodontal intervention, which might have concealed the genuine difference between the TG and CG in systemic and periodontal health indices. The low-intensity intervention in the CG was purposefully carried out to blind the patients to randomization, as they are part of the same BS support group. Another drawback was the inclusion of patients that underwent different types of BS, which might affect weight loss and resolution of obesity-related diseases. Future research should be done to determine whether periodontal therapy consistently impacts BS patients’ health.

A shift in the oral microbiome towards known periodontal pathogens is also observed in patients after BS [70,71,127]. Oral pathogens can affect the gut microbiome [45], and in BS patients, a high prevalence of small intestine bacteria overgrowth has been reported [128]. Future studies should aim towards proactive action, meaning to try to prevent the unfavorable shift of oral microbiome, ideally without applying chemical or pharmacological action. The use of probiotics, postbiotics, and paraprobiotic-based products for oral hygiene or in adjutancy to non-surgical periodontal therapy should be tested as aids in the reduction of the bacterial load in BS candidates [129,130,131,132]. The results of the RCT support the conduction of periodontal screening of BS candidates and intervention in periodontitis patients before the surgery [88]. Other precautions in BS patients that might affect dental health should be taken, such as mandatory endocrinological screening to assess the presence of osteoporosis, which is otherwise preventable with nutrition and supplementation [133]. An extensive study will be needed to address the possible positive outcome of periodontal therapy on the systemic health of BS patients.

## 5. Conclusions

Non-surgical periodontal therapy can both prevent further deterioration and improve periodontal health in BS patients 3 and 6 months after the surgery. Future studies should further explore the protentional beneficial effects of periodontal therapy on the systemic health of patients after BS.

## Figures and Tables

**Figure 1 jcm-11-06837-f001:**
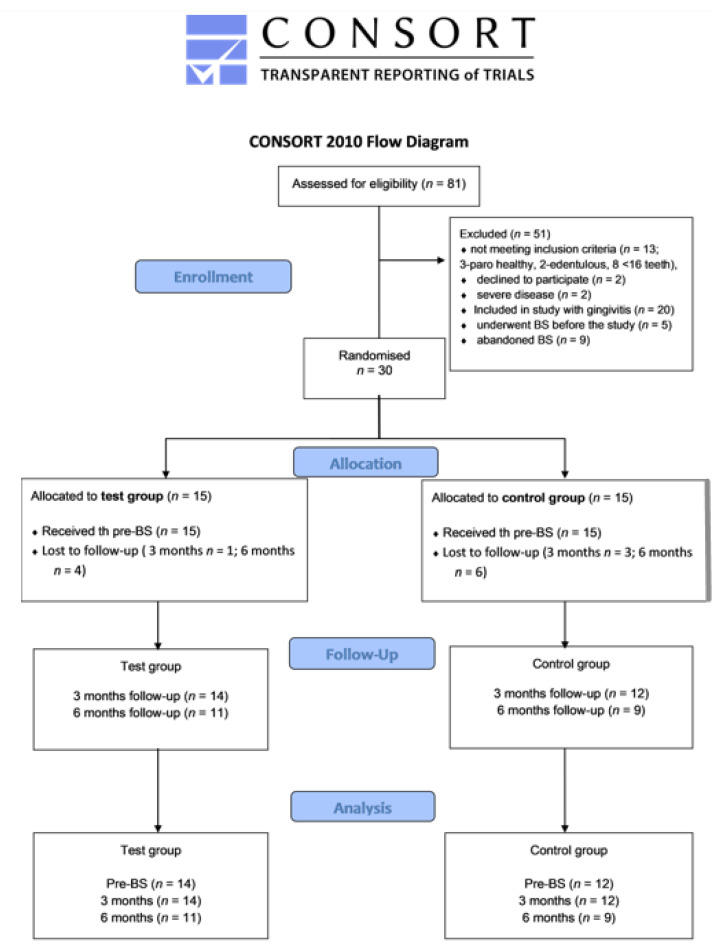
Flow diagram of patient recruitment and inclusion process. Pre-BS, baseline measurements.

**Table 1 jcm-11-06837-t001:** Baseline characteristics of the patients in the test and control groups.

Sociodemographic Characteristics	Test Group *n* = 14	Control Group *n* = 12	*p* Value
Age (years)	51.8 (7.6)	49.4 (11.1)	0.6
Woman	71% 10/14	75% 9/12	1
High level of education	7/14 (50% (23–77%))	5/12 (42% (15–72%))	0.7
Behavior habits			
Smokers	5/14 (36% (13–65%))	3/12 (25% (5–57%))	0.7
Alcohol use	1/14 (7% (0–34%))	1/12 (8% (0–38%))	1
Regular physical activity	5/14 (36% (13–65%))	6/12 (50% (23–71%))	0.7
Regular dental visits	9/14 (64% (35–87%))	7/12 (58% (28–85%))	1
Proper daily oral hygiene	6/14 (43% (18–71%))	4/12 (33% (10–65%))	0.7
Primary reason for bariatric surgery to improve health	6/14 (43% (18–71%))	8/12 (67% (35–90%))	0.3
**Obesity related parameters**	**Test group *n* = 14**	**Control group *n* = 12**	
BMI (kg/m^2^)	46.3 (5.4)	49.6 (7)	0.2
Waist circumference (cm)	131 (15)	135 (8.6)	0.4
EOSS			
1	2/14 (14% (2–43%))	1/12 (7% (0–34%))	1
2	2/14 (14% (2–43%))	4/12 (29% (8–58%))	
3	9/14 (64% (35–87%))	6/12 (43% (18–71%))	
4	1/14 (7% (0–34%))	1/12 (7% (0–34%))	
**Periodontal and dental parameters**	**Test group *n* = 14**	**Control group *n* = 12**	
Number of teeth missing	4 (3–9.5)	6.5 (2.5–10)	0.8
Denture present	2/14 (14% (2–43%))	2/12 (17% (2–48%)	1
Number of crowns	4 (0–5)	1 (0–5.5)	0.9
Number of pontics	0 (0–1)	0 (0–1)	0.6
Periodontitis stage			
I	4/14 (29% (8–58%))	3/12 (21% (5–51%))	1
II	4/14 (29% (8–58%))	4/12 (29% (8–58%))	
III	4/14 (29% (8–58%))	3/12 (21% (5–51%))	
IV	2/14 (14% (2–43%))	2/12 (14% (2–43%))	
Periodontitis grade			
A	2/14 (14% (2–43%))	2/12 (14% (2–43%))	1
B	7/14 (50% (23–77%))	5/12 (36% (13–65%))	
C	5/14 (36% (13–65%))	5/12 (36% (13–65%))	

BMI, body mass index; EOSS, Edmonton obesity staging system; Data presented as mean (standard deviation), or median (25–75 quartal), or prevalence (95% confidence interval) and number of patients.

**Table 2 jcm-11-06837-t002:** Weight loss after bariatric surgery.

Obesity Parameter	Pre-BS TG (*n* = 14)	Pre-BS CG (*n* = 12)	3 m TG (*n* = 14)	3 m CG (*n* = 12)	6 m TG (*n* = 11)	6 m CG (*n* = 9)	*p*-Value, ANOVA
BMI kg/m^2^	46.3 (5.4)	49.6 (7)	36 (5.6) #	39 (5) #	32 (6) #	35 (7) #	<0.001

BMI, Body mass index; Pre-BS, Measurements before bariatric surgery; 3 m, 3-month follow-up; 6 m, 6-month follow-up, TG, test group; CG, control group; #; test/control follow-up vs. Pre-BS, *t*-test, *p* < 0.05.

**Table 3 jcm-11-06837-t003:** Periodontal parameters before and after BS in test and control group.

Periodontal Parameters	Pre-BS TG (*n* = 14)	Pre-BS CG (*n* = 12)	3 m TG (*n* = 14)	3 m CG (*n* = 12)	6 m TG (*n* = 11)	6 m CG (*n* = 9)	*p*-Value, ANOVA
Bop %	32.5 (26.5–42.8)	37 (24.8–65.5)	30.5 (20–41)	52 (33–72)	30 (20–33) *	55 (51–70)	0.015
FMPI %	37 (28–45)	37 (21.5–77)	25 (16–32) #	38 (30–65)	25 (8–28) * #	60 (32–70)	0.0022
PPD mm	3.2 (2.7–3.3)	3.1 (2.9–3.5)	2.8 (2.4–3.1) *	3.4 (3.1–3.7)	2.8 (2.4–3) *	3.4 (3.2–3.8)	0.0015
%PPD > 4 mm	19.5 (18–23.5)	24 (14–27)	4.5 (3.3–9.5) * #	29 (11.5–33)	5 (1.5–10.5) * #	33 (22–36)	<0.001
CAL mm	1.3 (1–2)	1.2 (0.7–2)	1.3 (0.7–2.3)	1.4 (1–2.2)	1.5 (0.7–2.5)	1.9 (0.7–2.9)	0.9
REC mm	1.5 (1.2–1.5)	1.2 (1–1.9)	1.3 (1.2–1.5)	1 (1–2)	1.3 (1.1–1.5)	1.5 (1–2)	0.9
OHIP-14	6 (3–9.5)	7 (6–12)	3.5 (2–5.75)	6.5 (4–9.5)	4 (2–9)	15 (4–17)	0.44
Good oral hygiene practices	43% (17–81%)	33% (10–65%)	79% (49–95%)	58% (28–85%)	82% (48–98%)	67% (30–93%)	0.09 (Fisher’s exact test)

Pre-BS, Measurements before bariatric surgery; 3 m, 3-month follow-up; 6 m, 6-month follow-up; TG, test group; CG, control group; BOP, bleeding on probing; FMPI, full-mouth plaque index; PPD, periodontal probing depth; REC, recession; CAL, clinical attachment loss; data shown as median (25–75 quartal) or prevalence (95% confidence interval); *, test vs. control group at the same follow-up, U-test, *p* < 0.05; #, test/control group follow-up vs. Pre-BS, U-test, *p* < 0.05;.

**Table 4 jcm-11-06837-t004:** Obesity-related parameters and comorbidities before and after bariatric surgery.

Obesity-Related Parameters/Diseases	Pre-BS TG (*n* = 14)	Pre-BS CG (*n* = 12)	3 m TG (*n* = 14)	3 m CG (*n* = 12)	6 m TG (*n* = 11)	6 m CG (*n* = 9)	*p*-Value, Fisher’s Exact Test
**MS**	12/14 (86% (57–98%))	9/12 (75% (43–95%))	6/14 (43% (18–71%)) *	8/12 (67% (35–90%))	3/11 (27% (6–61%)) #	5/9 (56% (21–86%))	0.036
**Diabetes**	7/14 (50% (23–77%))	9/12 (75% (43–95%))	2/14 (14% (2–43%))	5/12 (42% (15–72%))	1/11 (9% (0–41%)) #	3/9 (33% (7–70%)) #	0.008
**High cholesterol**	12/14 (86% (57–98%))	10/12 (83% (52–98%))	6/14 (43% (18–71%)) *	7/12 (58% (28–85%))	4/11 (36% (11–69%)) #	6/9 (67% (30–93%))	0.039
**High triglyceride**	7/14 (50% (23–77%))	7/12 (58% (28–85%))	4/14 (29% (8–58%))	6/12 (50% (21–79%))	2/11 (18% (2–52%))	2/9 (22% (3–60%))	0.21
**Hypertension**	11/14 (79% (49–95%))	7/12 (58% (28–85%))	7/14 (50% (23–77%))	6/12 (50% (21–79%))	3/11 (27% (6–61%)) #	4/9 (44% (14–79%))	0.21
**Sleep apnea**	11/14 (79% (49–95%))	7/12 (58% (28–85%))	5/14 (36% (13–65%))	3/12 (25% (5–57%))	1/11 (9% (0–41%)) #	1/9 (11% (0–48%)) #	0.0014
**Joint and muscle problems**	10/14 (71% (43–92%))	7/12 (58% (28–85%))	6/14 (43% (18–71%))	7/12 (58% (28–85%))	4/11 (36% (11–69%)) #	4/9 (44% (14–79%))	0.31
**Depression**	2/14 (14% (2–43%))	2/12 (17% (2–48%))	1/14 (7% (0–34%))	3/12 (25% (5–57%))	2/11 (18% (2–52%))	2/9 (22% (3–60%))	0.87
**GERD**	8/14 (57% (29–82%))	7/12 (58% (28–85%))	6/14 (43% (18–71%))	5/12 (42% (15–72%))	3/11 (27% (6–61%))	3/9 (33% (7–70%))	0.62
**Habits**	**Pre-BS TG (*n* = 14)**	**Pre-BS CG (*n* = 12)**	**3 m TG (*n* = 14)**	**3 m CG (*n* = 12)**	**6 m TG (*n* = 11)**	**6 m CG (*n* = 9)**	***p*-value**
**Smoking**	5/14 (36% (13–65%))	3/12 (25% (5–57%))	1/14 (7% (0–34%))	1/12 (8% (0–38%))	1/11 (9% (0–41%))	2/9 (22% (3–60%))	0.36
**Alcohol consumption**	1/14 (7% (0–34%))	1/12 (8% (0–38%))	0/14 (0% (0–23%))	0/12 (0% (0–26%))	0/11 (0% (0–28%))	0/9 (0 (0–34%))	0.92
**Regular physical activity**	5/14 (36% (13–65%))	6/12 (50% (23–71%))	8/14 (57% (29–82%))	9/12 (58% (28–85%))	9/11 (82% (48–98%))	6/9 (67% (30–93%))	0.25

Pre-BS, Measurements before bariatric surgery; 3 m, 3-month follow-up; 6 m, 6-month follow-up; TG, test group; CG, control group; MS, metabolic syndrome; GERD, gastroesophageal reflux disease; *, test/control group at 3-month follow-up vs. Pre-BS, Fisher’s exact test, *p* < 0.05; #, test/control group at 6-month follow-up vs. Pre-BS, Fisher’s exact test, *p* < 0.05.

**Table 5 jcm-11-06837-t005:** Biochemical parameters of bariatric surgery patients before and 6 months after the surgery.

Biomarker	Pre-BS TG (*n* = 11)	Pre-BS CG (*n* = 9)	6 m TG (*n* = 11)	*p* Value Pre-BS vs. 6 m TG	6 m CG (*n* = 9)	*p-*Value, Pre-BS vs. 6 m CG	*p* Value, TG vs. CG at 6 m
CRP (mg/L)	12 (8–12.5)	16 (11–21)	3 (2.3–6.3)	0.018 *	4.5 (2.8–9.3)	0.028 *	0.46
AST (ukat/L)	0.4 (0.36–0.7)	0.41 (0.28–0.47)	0.47 (0.38–0.76)	0.9	0.4 (0.37–0.48)	0.62	0.38
ALT (ukat/L)	0.63 (0.47–0.79)	0.54 (0.42–0.63)	0.5 (0.35–0.60)	0.29	0.55 (0.42–0.73)	0.74	0.74
GGT (ukat/L)	0.34 (0.29–0.59)	0.35 (0.29–0.38)	0.27 (0.22–0.37)	0.3	0.29 (0.23–0.35)	0.6	0.85
HDL (mmol/L)	1.1 (1.1–1.4)	1.2 (1–1.3)	1.5 (1.2–1.7)	0.4	1.1 (1–1.6)	0.72	0.72
LDL (mmol/L)	3.6 (3–4)	3 (2.9–3.7)	2.2 (1.8–3.6)	0.31	2.7 (2.4–3.3)	0.72	0.44
Triglyceride (mmol/L)	1.5 (1.2–1.7)	1.9 (1.5–2.3)	1.2 (1.1–1.4)	0.31	1 (0.9–1.5)	0.1	0.44
Albumin (g/L)	45 (45–47)	46 (42.3–48)	42 (37–44)	0.005 *	44 (41–46)	0.32	0.13
Glucose (mmol/L)	5.3 (4.6–5.5)	6.6 (5.1–7.7)	4.7 (4.4–5)	0.09	5.2 (4.6–5.5)	0.23	0.19
HBA1c (%)	5.7 (5.3–6)	6.1 (5.1–6.8)	5.4 (5.3–5.5)	0.22	5.3 (5–5.7)	0.54	0.69

CRP, C reactive protein; AST, aspartate transaminase; ALT, alanine transaminase; GGT, gamma-glutamyl transpeptidase; HDL, high-density lipoprotein; LDL, low-density lipoprotein; HBA1c, glycated hemoglobin; Pre-BS, Measurements before bariatric surgery; 6 m, 6-month follow-up; TG, test group; CG, control group; *, U-test2, *p* < 0.05.

**Table 6 jcm-11-06837-t006:** Dietary patterns and gastric complications after bariatric surgery.

3 Months Post BS	Data (*n* = 26)
5–6 meals a day	25/26 (96% (79–100%))
Liquid diet after BS (weeks)	4.4 (1)
Soft food diet (weeks)	3 (1.8)
Solid food diet on 3-m control (yes %)	25/26 (96% (79–100%))
Taking supplements (yes %)	25/26 (96% (79–100%))
Vomiting at least 1xweek	10/26 (38% (17–59%))
How long did patients vomit	3.5 (1.2) weeks after BS
Proton pump inhibitors	26/26 (100% (86–100%))

BS, bariatric surgery.

## Data Availability

All data from the study are available on request from the corresponding author.

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
