# Peer review of "Periodontal Therapy in Bariatric Surgery Patients with Periodontitis: Randomized Control Clinical Trial"

_jcm, 2022, doi:10.3390/jcm11226837_

Round 1

Reviewer 1 Report

Review report on a manuscript: Periodontal therapy in bariatric surgery patients with periodontitis: randomized control clinical trial 

The present randomized, controlled, parallel-group, and double-blinded study might be of potential clinical interest since it presents data of nonsurgical periodontal therapy before bariatric surgery and its impact on the periodontal and systemic health parameters after the surgery. The results of the present study may be of great interest to clinicians since they may raise the awareness of the importance of periodontal health in BS patients. However, certain minor improvement of the article must be made, and the manuscript has to be re-worked. Please find the comments bellow.

1.     Title:

The title is representative of the aim of the study.

2.     Abstract: Abstract is well written.

       3. Background

On the page 2, line 61, you stated: “Some studies show that periodontal health deteriorates after BS.” Please provide more details on this topic, i.e. what was done in those studies and what were the main findings. 

4.     Material and methods

In section Study design you stated that CG subjects received low-intensive supragingival plaque removal with a mechanical brush and professional toothpaste (page 3, line 100). 

Please clarify if any additional tooth cleaning was between the baseline, 3m, and 6m visits. How the CG subjects were treated? Were the checkup visits organized more frequently than in TG? According to the provided information it may be assumed that was not the case. Please provide more detailed protocol and justify this treatment approach.

In section Patient recruitment you stated that all 30 patients underwent BS and periodontal therapy (page 3, line 143). However, only TG subject underwent periodontal therapy, while CG subject received only supragingival biofilm removal. Please, rephrase this sentence and clarify the periodontal treatment protocol for each study group.

5.     Results are presented well.

6.     Discussion 

Data on previously published studies on this topic are missing. Please provide more information what was already done in this field, and compare the results with the results of the present study.

On the page 11, line 329 you wrote: “The beneficial effects of periodontal therapy in TG were quite clear”. Please, rephrase this statement since only supragingival biofilm removal was done. 

Other results are well discussed. The limitations of the study, such as the modest sample size and the short follow up period, are discussed.

Even though the study showed that patients in TG tended to have higher resolution of MS components than patients in CG, the authors are encouraged to conduct future research to determine whether periodontal therapy has a consistent long-term positive impact on BS patients’ health. 

Author Response

Please see the attachment with the response to all reviewers.

Reviewer 2 Report

Manuscript of interest to the dental sector.

Abstract, to better highlight the statistical data

Keyword; add more specifications, these are few

Introduction The reference and description of the new classification of periodontal disease, related to systemic pathologies, is missing

Materials and methods; classify patients according to the new classification and enter how the sample size was calculated.

Results following the modified file, the results are not easily interpretable by ordinary readers who are not experts in scientific research, they make them more usable and highlight the requests more

Discussion, being an open-mouthed study, insert as future objectives, the use of a gel, toothpastes and mousse based on postbiotics and probiotics to reduce the bacterial load as already studied by the research group of Prof Scribante, in order to have a proactive approach and reduce the chemical pharmacological action over time both before, during and after bariatric surgery

Conclusion, rephrase it by adding proactive action

Author Response

(The authors gave the same response as above.)

Reviewer 3 Report

I congratulate the authors for conducting this important randomized trial. I have some minor comments.

-Please use the long forms of BOP and PPD in the abstract at first mention.
-Please discuss whether periodontal disease is just a bystander of the adverse effects of obesity-related chronic inflammation or whether it has roles in perpetuating chronic inflammation in obesity and metabolic syndrome. While periodontitis is generally regarded as a denominator of poor overall health, recent insights suggest more of a contributing role of periodontitis to disease pathogenesis. Please discuss these assumptions in light of the literature.
-Please discuss why the study could not reach a predefined target sample size (70 participants). Also, please clarify the confusion regarding the study completion date. It is noted as November 30, 2021, on clinicaltrials.gov.
-Adding p-values as a separate column for baseline comparison could be beneficial for table-1.
-Please consider adding p-values for the comparison of the baseline and third month to the table-4.
-Please discuss the possible reasons for the lack of effect on the OHRQoL with intervention further in the discussion.

Author Response

(The authors gave the same response as above.)

Round 2

Reviewer 2 Report

the manuscript has been correctly revised

Author Response

Point 1: the manuscript has been correctly revised, it can be published

Response 1: Thank you for your kind words.